# The Role of Biomarkers in the Management of Colorectal Liver Metastases

**DOI:** 10.3390/cancers14194602

**Published:** 2022-09-22

**Authors:** Daniel Brock Hewitt, Zachary J. Brown, Timothy M. Pawlik

**Affiliations:** Division of Surgical Oncology, Department of Surgery, The Ohio State University Wexner Medical Center, Columbus, OH 43210, USA

**Keywords:** colorectal cancer, colorectal liver metastasis, biomarker, ctDNA, radiomics

## Abstract

**Simple Summary:**

Colorectal cancer remains one of the most significant sources of cancer-related morbidity and mortality worldwide. The liver is the most common site of metastatic spread. Multiple modalities exist to manage and potentially cure patients with metastatic colorectal cancer. However, reliable biomarkers to assist with clinical decision-making are limited. Recent advances in genomic sequencing technology have greatly expanded our knowledge of colorectal cancer carcinogenesis and significantly reduced the cost and timing of the investigation. In this article, we discuss the current utility of biomarkers in the management of colorectal cancer liver metastases.

**Abstract:**

Surgical management combined with improved systemic therapies have extended 5-year overall survival beyond 50% among patients with colorectal liver metastases (CRLM). Furthermore, a multitude of liver-directed therapies has improved local disease control for patients with unresectable CRLM. Unfortunately, a significant portion of patients treated with curative-intent hepatectomy develops disease recurrence. Traditional markers fail to risk-stratify and prognosticate patients with CRLM appropriately. Over the last few decades, advances in molecular sequencing technology have greatly expanded our knowledge of the pathophysiology and tumor microenvironment characteristics of CRLM. These investigations have revealed biomarkers with the potential to better inform management decisions in patients with CRLM. Actionable biomarkers such as RAS and BRAF mutations, microsatellite instability/mismatch repair status, and tumor mutational burden have been incorporated into national and societal guidelines. Other biomarkers, including circulating tumor DNA and radiomic features, are under active investigation to evaluate their clinical utility. Given the plethora of therapeutic modalities and lack of evidence on timing and sequence, reliable biomarkers are needed to assist clinicians with the development of patient-tailored management plans. In this review, we discuss the current evidence regarding biomarkers for patients with CRLM.

## 1. Introduction

Colorectal cancer (CRC) is a leading cause of cancer-related morbidity and mortality worldwide [1]. While the global incidence continues to rise, disproportionately so in low- and middle-income countries, advances in CRC screening and multidisciplinary therapy have significantly improved mortality rates, especially among patients living in highly developed nations where mortality rates have declined in recent years [2]. Despite these improvements, CRC remains the second leading cause of cancer-related mortality in the United States [3].

Metastatic liver disease develops in approximately 50% of patients with CRC [3]. While most cases of colorectal liver metastases (CRLM) develop metachronously after treatment for locoregional CRC, 20% to 34% of patients with CRC present with synchronous liver disease. Synchronous disease portends a worse prognosis as these patients more frequently have multiple liver lesions and bilobar disease versus patients with metachronous CRLM [4]. Unfortunately, 80% to 90% of patients with CRLM have an unresectable liver disease at presentation [5,6]. Five-year survival is significantly lower among patients not undergoing surgery [7]. Consequently, metastatic liver disease drives mortality in most patients with CRC.

The current standard of care therapy for patients with resectable CRLM includes both curative intent surgery and perioperative systemic therapy [8,9]. Refined surgical techniques combined with more efficient surgical devices have greatly reduced perioperative morbidity and mortality from liver surgery while improving resection rates [10]. Complete surgical resection with negative margins and adequate liver remnant significantly improves long-term outcomes for patients [9,11]. Recent meta-analyses reported a median 5-year survival of 38% to 71% for select patients with solitary liver metastases following resection with appropriate perioperative systemic therapy [12,13].

However, 50% of patients develop recurrent disease within two years after surgery [14]. While some of these patients are candidates for additional resection, many have an unresectable disease. For patients with unresectable CRLM, liver-directed therapies provide local disease control and, in certain situations, may convert patients to resectable disease. Liver-directed therapies include hepatic artery infusion (HAI), arterial-directed embolization (i.e., radioembolization (Y-90)), ablation, or external beam radiation (i.e., conformal radiation therapy (CRT), stereotactic body radiation therapy (SBRT), and intensity-modulated radiation therapy (IMRT)) [8]. While surgery remains the gold standard therapy, questions remain over the choice and sequence of perioperative systemic therapy, as well as the necessity and timing of liver-directed therapies. In fact, even among expert liver surgeons, significant debate exists regarding the appropriate therapeutic strategy in more than half of the clinical cases proposed in a recent survey [15].

Advances in molecular sequencing technology (e.g., next-generation sequencing (NGS)) and computational data analytics have greatly improved our understanding of CRC pathophysiology, revealing genomic variants responsible for CRC carcinogenesis. These molecular biomarkers, combined with the emerging fields of radiomics and artificial intelligence, provide clinicians with potentially actionable information. Current treatment algorithms for patients with CRLM incorporate certain biomarkers, including RAS, BRAF, and mismatch repair (MMR) status. Other biomarkers require validation on their clinical utility and are actively being investigated in clinical trials. We herein review the current utility of biomarkers in the management of patients with CRLM.

## 2. Tumor Morphology

Tumor morphology provides significant prognostic information and guides clinical decision-making. The size, number of lesions, and proximity to vascular structures determine resectability, and margin-negative resection is an important factor associated with long-term survival. Furthermore, lesion size, depth from the periphery, and vessel proximity inform combined surgical/ablation approaches to provide definitive management and spare liver parenchyma in the case that future hepatectomy may be needed to clear recurrent disease [16,17]. Second, morphology-based scoring systems demonstrate a prognostic discriminatory ability for long-term survival outcomes (Table 1) [18,19,20,21]. An earlier scoring system developed by Fong et al. included the number of lesions (>1) and size of the largest lesion (>5 cm) among five clinical or morphologic factors and successfully prognosticated survival after hepatectomy [18]. A recent study by Sasaki et al. developed a Tumor Burden Score (TBS) using only the number of lesions and maximum tumor size based on similar studies that included patients with hepatocellular carcinoma [22]. External validation on two cohorts demonstrated accurate stratification of 5-year survival based on the TBS, with higher scores correlating with lower survival among patients undergoing hepatectomy for CRLM. Peng et al. applied the TBS to patients with unresectable CRLM and reported that the TBS successfully predicted conversion to resectable disease [23].

Radiographic response rates of CRLM while on systemic or liver-directed therapy have been correlated with improved disease-free and overall survival among patients with resectable and unresectable diseases [25]. Patients with a complete radiographic response (i.e., disappearing liver metastases) after therapy that remains undetectable after intraoperative ultrasound have a pathologic complete response (pCR) in 24% to 96% (median 77.5%) of cases [26]. Furthermore, patients with the unresectable disease who converted to resectable disease after therapy demonstrated improved survival compared with patients who did not demonstrate a radiographic response and remained unresectable [27]. Histological growth patterns of CRLM evaluated after resection also provide a prognostic value regarding overall survival [11,28].

In response to the growing amount of clinical information generated from the digitalization of healthcare, investigators have explored artificial intelligence (AI)-based techniques, such as machine learning. While not a biomarker itself, AI optimizes data utility by considering all available data in an unbiased and unsupervised manner with the ability to continuously improve predictions [29]. A novel machine learning approach was used to predict long-term outcomes in 1406 patients with CRLM that underwent hepatic resection [30]. The machine-learning generated model based on clinical and morphological tumor characteristics significantly outperformed traditional clinical risk scores for predicting 1-, 3-, and 5-year recurrence.

Tumor morphology has some significant limitations as a “biomarker”. Tumor morphology provides incomplete information on resectability. For example, specific criteria and thresholds to define resectability vary among institutions, sometimes based more on surgeon expertise than morphologic characteristics. In addition, tumor morphology offers little information to identify the 10% to 15% of patients with a resectable disease who undergo a resection yet develop early recurrence and cancer-related death [31]. While changes in tumor morphology based on serial imaging provide valuable information about the response to therapy during treatment, it poorly predicts response to systemic therapy prior to the initiation of treatment, let alone determine which systemic therapy may provide the best outcome for a particular patient.

## 3. Molecular Biomarkers

In 1988, Vogelstein et al. published a seminal report characterizing mutations related to KRAS, APC, and TP53 at various stages of CRC carcinogenesis, hypothesizing that invasive carcinoma develops from adenomatous polyps via the sequential acquisition of somatic mutations in multiple genes [32]. Since this foundational work, the development/invention of NGS technology has dramatically advanced the field of oncogenesis with techniques such as whole-genome sequencing, whole-exome sequencing, and targeted sequencing that have revealed a host of genomic alterations [33]. Figure 1 illustrates the interconnected relationships between multiple signaling cascades critical in CRC oncogenesis, highlighting two frequent mutation genes, the tumor suppressor gene TP53 (pathway A) and proto-oncogene PI3K (pathway B).

The CRC-liver metastatic cascade is a complex process wherein a subset of CRC cells acquires the capacity to evade the primary tumor, migrate through the extracellular matrix and neighboring tissue, intravasate, survive transit through the circulation, extravasate, and ultimately colonize the liver [35]. The molecular alterations required to complete this process create a biologically aggressive and phenotypically distinct disease entity.

Genomic alterations do not always correspond to predictable changes in biological activity [36]. Proteomics provides clarity to the molecular processes occurring in the gap between gene expression and disease phenotype. Recent proteomic profiling studies revealed unique protein and post-translational modifications present in metastatic CRC tumors, especially in proteins associated with the extracellular matrix, energy metabolism, and immune-cell-related migration [37,38,39]. In the following section, we discuss significant genomic, proteomic, and post-translational mutations in CRLM development.

### 3.1. Genomic Biomarkers

Genome-wide sequencing studies have demonstrated actionable mutations in one-third of patients with CRC metastasis (Figure 2) [40,41]. The current National Comprehensive Cancer Network (NCCN) guidelines recommend testing for RAS (KRAS and NRAS) and BRAF mutations and HER2 amplifications, individually or as part of an NGS panel, for all patients with metastatic CRC and universal MMR or microsatellite instability (MSI) testing in all newly diagnosed patients with CRC [8].

Alterations in the RAS proto-oncogene family, most notably KRAS, NRAS, and HRAS, result in unregulated cell proliferation via gain-of-function activity in the MAPK pathway. RAS mutations are found in up to 52% of patients with CRLM and have a high concordance between the primary tumor and CRLM [43,44]. Patients with RAS-mutated CRLM have significantly worse recurrence-free and overall survival versus individuals with wild-type RAS [45,46]. Due to the upregulation of the MAPK pathway, patients with RAS mutations quickly develop resistance to epidermal growth factor receptor (EGFR) antibody therapy [47,48]. As a result, anti-EGFR antibody agents, cetuximab and panitumumab, are only recommended in the treatment of KRAS and NRAS wild-type tumors [8]. Furthermore, RAS-mutated CRLM demonstrate more migratory or invasive biology, causing local tumor progression and a higher incidence of micrometastasis compared with wild-type RAS CRLM [49,50,51]. Consequently, these tumors have a narrower median negative margin (4 mm vs. 7 mm) and double the rate of positive surgical margins after hepatectomy [52]. Similar results are seen for smaller RAS-mutated CRLM treated with ablation therapy [34]. The appropriate negative margin distance for patients with CRLM remains a controversial topic [53]. Where patients with wild-type RAS CRLM benefited from negative surgical margins, patients with RAS-mutated CRLM had similarly poor outcomes between the R0 and R1 resection groups [54,55]. A recent analysis of 1843 patients with CRLM who underwent curative-intent surgery used AI-based analytics to determine the optimal surgical margin in KRAS-variant CRLM [56]. The AI model suggested an optimal margin width of 7 mm for KRAS-variant CRLM. Most of the associated prolongation of survival was seen with a 1 mm margin, with the extension from 1 mm to 7 mm contributing a smaller proportion of the improvement in survival. Ultimately, patients with RAS-mutated tumors should demonstrate disease stability on systemic therapy, with the absence of other poor prognostic factors, prior to attempting curative-intent surgical resection.

BRAF, another protein in the MAPK pathway, has emerged as a very poor prognostic indicator [57]. BRAF mutations occur in 5% of patients with CRLM. BRAF^V600E^ contains a substitution of valine for glutamic acid at codon 600 and is responsible for over 90% of these mutations [58]. Patients with BRAF-mutated CRC rarely present with isolated liver metastases. Furthermore, even in the small proportion of patients who present with resectable disease, median recurrence-free survival and overall survival after curative-intent hepatectomy is half compared to wild-type BRAF patients [59]. Similar to RAS-mutated CRC, patients with a BRAF mutation do not respond to anti-EGFR therapy unless administered with a BRAF inhibitor (i.e., encorafenib) [60]. Although data are limited to small studies, patients with non- BRAF^V600E^ mutations may have better outcomes compared with even wild-type BRAF patients [61].

HER2 amplification is a targetable variant in the MAPK pathway found in 2–3% of metastatic CRCs [62]. While well-studied in breast cancer, the low prevalence of HER2 amplification in CRC cases limits confident statements about the prognostic effect of HER2 amplification. Regarding targeted therapies, phase 2 trials support the use of a dual HER2 blockade in heavily pretreated, HER2-amplified metastatic CRC [63,64]. However, anti-HER2 therapy is only indicated in HER2-amplified tumors that are also RAS and BRAF wild-type.

Approximately 5% of patients with metastatic CRC harbor deficient DNA mismatch repair (dMMR), leading to MSI [34]. Alterations in the MMR system arise through germline mutations (i.e., Lynch syndrome) or sporadically from promoter hypermethylation and silencing of the MMR gene MLH1 [65]. Sporadic mutations are highly associated with BRAF^V600E^ mutations, present in one-third of dMMR patients, and have a worse prognosis versus patients with a germline dMMR [66]. In early-stage CRC, dMMR is associated with reduced metastatic potential and a favorable prognosis; however, dMMR status in metastatic CRC results in a worse prognosis compared with proficient MMR tumors [67]. Metastatic dMMR/MSI-H CRC responds well to immune checkpoint inhibitor (ICI) therapy, doubling the progression-free survival time compared with traditional chemotherapy (16.5 months vs. 8.2 months; HR 0.60, *p* = 0.0002) [68]. In addition, patients with metastatic CRC can have a high tumor mutation burden leading to MSI independent of the dMMR status. Patients with a high tumor mutational burden may also benefit from ICI therapy [69].

Unfortunately, most of the genomic variants identified in metastatic CRC lack a targeted therapy, including four of the five most mutated genes: APC, TP53, PIK3CA, and SMAD4. However, knowing the status of these mutations does provide prognostic information. Concomitant TP53 and RAS mutations have negative prognostic effects worse than either mutation alone. Specifically, patients with CRLM containing both mutations have significantly worse recurrence-free survival and overall survival [70]. A similar pattern emerges for patients with tumors containing both an APC and PIK3CA mutation. Together, these mutations have a synergistic effect on chemoresistance and, as a result, a worse prognosis [71,72]. Not surprisingly, tumors containing both a tumor suppressor mutation (TP53, APC) and an oncogene (KRAS, PIK3CA) are more biologically aggressive, leading to earlier disease recurrence and mortality.

### 3.2. Proteomic Biomarkers

Early proteomic studies on CRLM revealed diverse proteomic profiles demonstrating significant metastatic CRC tumor heterogeneity [73]. Current investigations remain in the preclinical exploratory phase, but these studies have identified promising targets for future research. In particular, studies comparing the proteome biology of primary CRC and CRLM have revealed distinct protein dysregulation profiles [37,38,74]. Fahrner et al. compared the matched proteomes of seven patients with CRLM and noted upregulated proteins in liver metastasis involving metabolic processes such as gluconeogenesis (pyruvate carboxylase) and fructose metabolism (fructose-bisphosphate aldolase B (ALDOB), fructose-1,6-bisphosphatase (1), as well as proteins linked to the complement cascade, indicating an active immune response) [37]. ALDOB upregulation indicates a poor prognosis and has been demonstrated to promote tumor progression and CRLM by facilitating the epithelial-mesenchymal transition [75]. Conversely, comparatively downregulated proteins are involved with cellular structural integrity/cell junction assembly (desmin, synemin, and filamin-C). In a separate study of eight patients with CRLM, several extracellular matrix components were significantly upregulated in CRLM compared with the primary tumor, most notably THBS1, which facilitates CRLM by promoting epithelial–mesenchymal transition [38]. Furthermore, THBS1 upregulation was associated with shorter overall survival. CD11b and ITGA2 are two additional proteins with a role in promoting the epithelial–mesenchymal transition that are overexpressed in CRLM [76]. In addition, Ku et al. demonstrated that the filamin A-interacting protein 1-like (FILIP1L) and plasminogen were dysregulated in CRLM compared with primary CRC [74]. FILIP1L overexpression inhibits WNT signaling and decreases metastatic behavior in CRC [77]. Ku et al. reported the downregulation of FILIP1L in CRLM samples. In contrast, plasminogen has been noted to be overexpressed in CRLM samples. Overexpression of the plasminogen activating system in CRLM has been associated with worse overall and cancer-specific survival [78].

The extracellular matrix (ECM) provides physical scaffolding for cells and plays a vital role in biochemical signaling [79]. Alterations in the composition of the ECM are linked to pathological conditions, including carcinogenesis and metastasis. The ECM in cancer patients is often disorganized and characterized by the upregulation of many components (e.g., collagen) compared with normal tissue [80]. Naba et al. performed the first proteomic study of the ECM composition in patients with CRLM, demonstrating that the ECM composition of CRLM resembled the ECM of primary CRC more so than a normal liver [81]. Furthermore, seven proteins were uniquely associated with metastases, including TIMP1 (Tissue Inhibitor of Metalloproteinase-1), a protein that induces pro-invasive ECM remodeling when upregulated and has been associated with poor progression-free survival in patients with CRLM [82]. In addition, TIMP1 may have utility as a non-invasive biomarker for preoperative risk stratification in patients with CRLM [82].

Protein post-translational modifications (PTMs) such as crosslinking, acetylation, ubiquitination, methylation, glycosylation, and citrullination can occur on histone and nonhistone proteins. These modifications often contribute to protein degradation to maintain normal physiological homeostasis; however, the deregulation of PTMs supports carcinogenesis and the development of metastasis. Shen et al. constructed a complete atlas of differentially expressed acetylated proteins in primary CRC and paired CRLM [83]. The authors identified 71 acetylated sites on 55 proteins in CRLM. These proteins were found primarily in the cytoplasm and associated with a broad range of biological processes, including metabolic pathways and carbon metabolism. Of the acetylated nonhistones, TPM2 and K152Ac were the most upregulated, and ADH1B K331Ac was the most downregulated. In another PTM study, Yuzhalin et al. demonstrated that peptidylarginine deiminase 4 (PAD4)-dependent protein citrullination plays an integral role in the development of CRLM [84]. Citrullination is characterized by a post-translational conversion of arginine residues to citrulline. In the study, CRLM exhibited significantly higher levels of PAD4 and citrullination compared to both healthy livers and the primary CRC. Furthermore, citrullination may facilitate CRLM by promoting the epithelial–mesenchymal transition since citrullination of collagen type 1 in the ECM promoted greater adhesion, decreased migration of CRC cells, and increased expression of epithelial markers.

Recently, proteomics has played a central role in the multiomic evaluation of CRLM. Multiomics provides a more comprehensive evaluation of biological processes by integrating data generated from multiple omic analyses (e.g., proteomic, transcriptomic, and genomic) [85]. A recent, large multiomic study integrated genomics, proteomics, and phosphoproteomics to provide a global evaluation of 146 patients with CRC, including 43 patients with CRLM [86]. CRLM were genomically similar to primary CRC but exhibited significant heterogeneity at the proteomic and post-translational level. In addition, in vivo drug testing suggested that phosphoproteomic profiling may more accurately predict drug response to kinase inhibitors than the presence of genomic mutations. Multiomic studies have also demonstrated a correlation with survival [87,88]. Two separate studies by Ma et al. reported that the mutated peptide number had prognostic value and that somatic variants demonstrated corresponding dysregulated protein abundance and biologic function. Relevant variants identified in CRLM included UQCR5, FDFT1, MYH9, and CCT6A.

### 3.3. Liquid Biopsy and ctDNA

A core tumor biopsy represents the current gold standard for obtaining tissue samples for molecular analyses. Core biopsies can be fraught with technical complications and fail to accurately capture complete tumor spatial heterogeneity and tumor evolution. [89,90]. Liquid biopsy collects body fluid (e.g., blood, urine, saliva) for molecular evaluation and may overcome many of the limitations associated with core biopsy [91]. Patients with solid tumors release isolated tumor DNA fragments into the blood after tumor lysis or apoptosis, called circulating tumor DNA (ctDNA). ctDNA is distinct from normal circulating-free DNA (cfDNA) and contains specific pathologic genetic variants [92]. CRCs tend to shed high amounts of ctDNA relative to other solid tumors, making them an ideal candidate for further investigation [93].

Investigation into the clinical utility of ctDNA as a biomarker for patients with CRC is ongoing (Table 2). Surgical resection of CRLM remains the standard of care, even for patients who have a clinical complete response (cCR) to neoadjuvant therapy, as available biomarkers cannot accurately identify patients who have achieved a pCR [26]. Studies specifically evaluating ctDNA in the neoadjuvant setting among patients with CRLM are limited; however, in retrospective studies of patients with stage II-III CRC, the ctDNA status following neoadjuvant therapy was associated with the pCR status [94,95], disease-free survival, and overall survival [96,97]. Furthermore, sequential sampling before, during, and after therapy allows for the real-time monitoring of the treatment response and disease evolution. Detection of ctDNA during neoadjuvant therapy correlated with primary tumor regression and shorter metastasis-free survival [98].

In retrospective studies, ctDNA demonstrates a prognostic value after hepatectomy for patients with CRLM. In a single institution study of 63 patients with CRLM, 42 (67%) had ctDNA present after hepatectomy. These patients had significantly worse overall survival, especially when multiple gene mutations were detected [100]. Similar results were demonstrated by Tie et al. in a cohort of 49 patients with resectable CRLM who underwent curative-intent hepatectomy [101]. The 11 patients with positive ctDNA after resection had lower recurrence-free survival and overall survival. However, ctDNA clearance was achieved with adjuvant therapy in three patients, two of whom remained disease free. Patients with positive ctDNA at the completion of therapy (surgery +/− adjuvant chemotherapy) had a 5-year recurrence-free survival of 0% versus 75.6% for patients with undetectable ctDNA after therapy.

Among patients with unresectable metastatic CRC, ctDNA-derived mutational analyses were nearly 100% concordant with the solid tissue biopsies for the detection of multiple clinically relevant mutations, such as the BRAF^V600E^, KRAS, and NRAS mutations [102,103]. Of note, intra-tumor heterogeneity, the treatment effect, or a low disease burden were responsible for the few discordant results. Furthermore, recent phase II clinical trials demonstrate the efficacy of ctDNA-guided rechallenge therapy with anti-EGFR drugs in the RAS wild-type unresectable metastatic CRC. Sartore-Bianchi et al. evaluated 27 patients who were ctDNA negative for the RAS/BRAF/EGFR mutations. Rechallenge therapy with panitumumab was associated with disease control in 59% of patients with a median progression-free survival of 16 weeks [104]. A separate study by Martinelli et al. included 77 patients with unresectable metastatic CRC that failed second-line therapy after acquiring resistance to first-line chemotherapy plus anti-EGFR drugs. Among the 48 patients with RAS/BRAF wild-type ctDNA, rechallenge therapy with cetuximab plus avelumab improved the median overall survival [105].

Despite the growing evidence supporting the utility of ctDNA, its widespread incorporation into clinical decision-making algorithms has yet to occur due to some notable limitations [106]. While patients with CRCs tend to shed higher levels of ctDNA compared with other cancer types, the concentration of ctDNA is still low, especially earlier in the disease process and compared with cfDNA. Highly sensitive tests are required to detect ctDNA accurately. Furthermore, the ratio of ctDNA to cfDNA (ctDNA fraction) significantly influences assay sensitivity and specificity. As a result, variation in the ctDNA fraction leads to unclear thresholds for the limit of detection [89,92]. Finally, currently available evidence supporting the use of ctDNA is overwhelmingly retrospective. Results from active randomized controlled trials will hopefully provide clarity on the clinical utility of ctDNA in patients with CRLM.

## 4. Radiomics

Radiomics refers to the quantitative analysis of textual features in medical images using data-characterization algorithms [107,108]. In theory, by examining the distinct radiomic features (e.g., shape, intensity, sphericity) in a region of interest, radiomic-based approaches extract biological data directly from the medical images, providing timely diagnostic and prognostic information while avoiding the cost and morbidity of an invasive biopsy [109].

Lubner et al. reported one of the first studies evaluating the clinical utility of radiomic features in patients with CRLM [110] The authors examined CT scans in 77 patients with a single untreated CRLM and noted an association between the radiomic features and tumor grade, KRAS mutation, and overall survival. Radiomic-based approaches have also been used to predict the effectiveness of therapy. In two recent studies, radiomic signatures successfully predicted treatment sensitivity to EGFR-targeted therapy [111]. and FOLFIRI + bevacizumab [112], outperforming known biomarkers such as tumor shrinkage as determined by RECIST. Furthermore, radiomic signatures were significantly associated with overall survival in both studies. In another study, a CT-based radiomics model outperformed a clinical model to detect local tumor progression in 31 lesions after thermal ablation [113]. In addition to predicting the therapy response, radiomics may improve the diagnostic accuracy of chemotherapy-associated liver injury (CALI). In a retrospective study of 78 consecutive patients with CRLM that received preoperative chemotherapy followed by liver resection, 66 demonstrated some degree of CALI on the final pathology [114]. Multiple radiomic features were associated with CALI, and, combined with relevant clinical data, radiomic signatures significantly improved the diagnostic accuracy of CALI versus traditional models.

AI techniques may optimize the predictions of radiomic-based analyses given the abundance of data extracted from medical images during the process. In a retrospective study of 91 patients with CRC, machine learning-derived models were used to predict the occurrence of metachronous CRLM [115]. The model, trained on only radiomic features, significantly outperformed the model that used only clinical features (AUC 85% [95%CI 85–87%] vs. 71% [95%CI 69–72%]). In another retrospective study, a deep learning-based radiomics model better-predicted response to chemotherapy in 192 patients with CRLM compared with the traditional classifier-based radiomics model [116]. In another study of 199 patients with CRC and 550 small hypoattenuating hepatic nodules (<1 cm), convolutional neural networks (CNNs) characterized lesions as benign or malignant with similar accuracy as expert radiologists, yet with greater diagnostic confidence (the CNN had a lower rate of nodules reported with low confidence, 19.6 vs. 31.4 (expert average); *p* < 0.0001).

Although radiomics has shown potential as a non-invasive biomarker for patients with CRLM, several challenges should be mentioned. While radiomic features have been standardized [117], significant heterogeneity exists between the number and order of feature analysis, and nearly all institutions use a different software application [109], Furthermore, when reported, radiomic feature thresholds vary between studies, limiting reproducibility and generalizability. Few radiomic studies evaluating CRLM have a prospective design or include a validation dataset. Despite these limitations, radiomic features demonstrate diagnostic and prognostic value for patients with CRLM in a timely and non-invasive manner.

## 5. Conclusions

CRLM continues to be a major source of morbidity and mortality for patients, despite a multitude of therapeutic modalities. However, NGS techniques and advances in mass spectrometry have improved our understanding of the genomic, proteomic, and post-translational variants responsible for CRLM development, revealing many potential biomarkers. Furthermore, the application of AI technology and, more specifically, machine learning algorithms provides an unbiased method to integrate the immense amount of molecular and radiomic data being generated to assist clinicians with developing personalized treatment plans. The many active trials evaluating the clinical utility of ctDNA as a less invasive biomarker are expected to provide clarity on the diagnostic and prognostic role of ctDNA in the management of patients with CRLM. While much work remains to optimize treatment plans, recent advances in the field provide hope for improving outcomes for patients with CRLM.

## Figures and Tables

**Figure 1 cancers-14-04602-f001:**
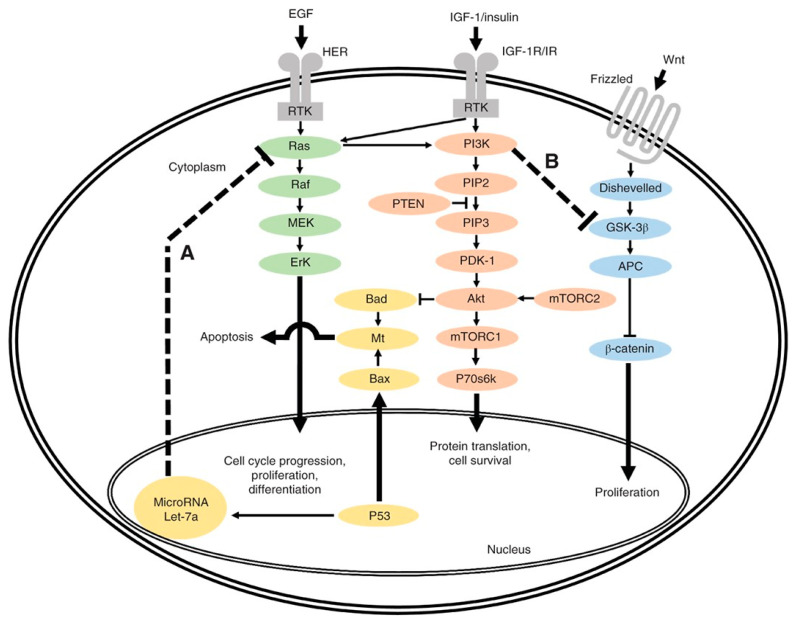
Overview of interlinked cellular signaling pathways involved in the proliferation and progression of colorectal cancer. In pathway A, TP53 normally inhibits activated RAS through lethal (Let) 7a23. However, Let-7a is not able to regulate activated RAS if TP53 is mutated. In pathway B, overactive phosphoinositide 3-kinase (PI3K), an oncogene, inhibits glycogen synthase kinase (GSK) 3β24, leading to β-catenin accumulation. EGF, epidermal growth factor; HER, human epidermal growth factor receptor; RTK, receptor tyrosine kinase; MEK, mitogen-activated protein kinase; ErK, extracellular signal-regulated kinase; Mt, mitochondria; IGF-1, insulin-like growth factor 1; IGF-1R/IR, IGF-1 receptor/insulin receptor; PIP2, phosphatidylinositol 4,5-bisphosphate; PTEN, phosphatase and tensin homologue; PIP3, phosphatidylinositol (3,4,5)-trisphosphate; PDK-1, phosphoinositide-dependent protein kinase 1; mTORC1/2, mammalian target of rapamycin complex 1/2; P70s6k, P70s6 kinase; APC, adenomatous polyposis coli. Used with permission [34].

**Figure 2 cancers-14-04602-f002:**
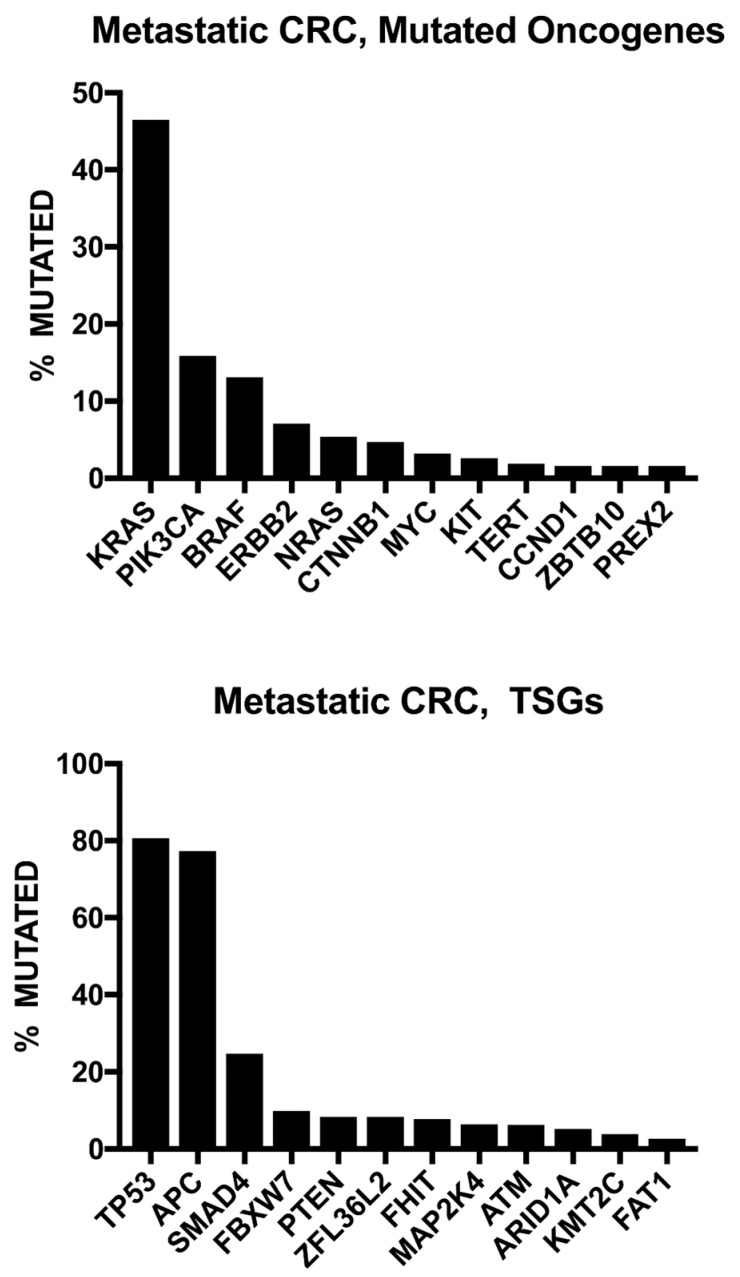
Frequent gene alterations based on wide-genome sequencing of 372 metastatic colorectal cancer (CRC) patients. Note: TSGs: tumor suppressor genes. Open Access Citation: Testa U, Castelli G, Pelosi E. Genetic Alterations of Metastatic Colorectal Cancer. *Biomedicines*. 2020; 8(10):414. https://doi.org/10.3390/biomedicines8100414 [42].

**Table 1 cancers-14-04602-t001:** Clinical risk scores.

Study	CRS Criteria (1 Point for 1 Risk Factor)	Risk Groups
Fong [18]	1. Largest liver metastasis > 5 cm 2. Disease-free interval < 12 months 3. Number of liver metastases > 1 4. Lymph node-positive primary tumor 5. CEA > 200 ng/mL	Low 0–2 pts High 3–5 pts
Nordlinger [19]	1. Age > 60 years 2. Serosal invasion of primary tumor (≥pT3) 3. Lymph node-positive primary tumor (pN1) 4. Disease-free interval < 24 months 5. Number of liver metastases > 3 6. Largest liver metastasis > 5 cm	Low 0–2 pts Intermediate 3–4 pts High 5–6 pts
Nagashima [20]	1. Serosal invasion of primary tumor (≥pT3) 2. Lymph node-positive primary tumor (pN1) 3. Number of liver metastases ≥ 2 4. Largest liver metastasis ≥ 5 cm 5. Resectable extrahepatic metastases	Low 0–1 pts Intermediate 2–3 pts High ≥4 pts
Konopke [21]	1. Number of liver metastases ≥ 4 2. CEA ≥ 200 ng/mL 3. Synchronous liver metastases	Low 0 pts Intermediate 1 pt High ≥2 pts

Open access citation: Wimmer, K., Schwarz, C., Szabo, C. et al. Impact of Neoadjuvant Chemotherapy on Clinical Risk Scores and Survival in Patients with Colorectal Liver Metastases. *Ann Surg Oncol* 24, 236–243 (2017). https://doi.org/10.1245/s10434-016-5615-3 [24].

**Table 2 cancers-14-04602-t002:** Notable clinical trials of ctDNA in patients with metastatic colorectal cancer.

Study (Code Identifiers) Location	Trial DesignStatus	Estimated Enrolment (N pts) ctDNA Analysis	Main Characteristics and Inclusion Criteria
(NCT03844620) USA	Phase II Recruiting	100 NA	Pts clinically eligible for either regorafenib or trifluridin-tipiracil Pts will continue treatment beyond 1st cycle depending on ctDNA results
(NCT04831528) China	Phase II Not yet recruiting	100 NA	Pts must have failed after first-line treatment containing cetuximab Individualized second-line targeted therapy based on ctDNA analysis
FOLICOLOR (NCT04735900) International	NA Recruiting	60 NPY Methylation	Unresectable metastatic disease Identification of PD by NPY Methylation in liquid biopsies To assess response and progression to first-line FOLFOX/FOLFIRI treatment on liquid biopsy
NCT04509635 China	Phase III Not yet recruiting	50 NA	*RAS* wt on ctDNA Non-resectable liver metastases candidate for anti-EGFR rechallenge based on ctDNA results
LIBImAb (NCT04776655) Italy	Phase III Not yet recruiting	280 *KRAS*, *NRAS*, and in *BRAF^V600^* status assessment using the Idylla system (Biocartis)	*RAS/BRAF* wt on solid tumor biopsy but with *RAS* mutant at liquid biopsy To compare di efficacy of FOLFIRI + Cetuximab or Bevacizumab in tissue wt but liquid mutant *RAS* mCRC
NCT04224415 China	Phase II Not yet recruiting	35 *RAS*/*BRAF* status assessment	First-line therapy of FOLFOX/FOLFIRI/FOLFOXIRI + Cetuximab effectively and the PFS is not less than 6 months ≥4 months after the last time treated with Cetuximab *RAS/BRAF* wt on ctDNA
PARERE (NCT04787341) Italy	Phase II Recruiting	214 IdyllaTM ct*KRAS*-*NRAS*-*BRAF* Mutation Test	*RAS* and *BRAF* wt status of primary CRC or related metastasis *RAS* and *BRAF* wt ctDNA at the time of screening Previous first-line anti-EGFR-containing therapy with at least a PR or SD ≥ 6 months; ≥4 months elapsed between the end of first-line anti-EGFR administration and screening; ≥1 line of therapy between the end of first-line anti-EGFR administration and screening
NCT04775862 Saudi Arabia	Phase II Recruiting	60 *RAS* status assessment	Baseline must be *RAS/BRAF* wt on solid tumor tissue *RAS* wt on ctDNA Tumor burden with <4 organ involvement
NCT03992456 USA	Phase II Recruiting	120 Guardant360 assay	*RAS* and *BRAF* wt on tumor tissue taken from primary or metastatic site PD after treatment with an anti-EGFR monoclonal antibody for at least 4 months ≥90 days from the last anti-EGFR treatment *BRAF, EGFR, ERBB2, RAS, MET* wt highest allele frequency reported for any gene mutation < 2%

Open access citation: Mauri, G., Vitiello, P.P., Sogari, A. et al. Liquid biopsies to monitor and direct cancer treatment in colorectal cancer. *Br**. J. Cancer* (2022). https://doi.org/10.1038/s41416-022-01769-8 [99].

## Data Availability

Not applicable.

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
