# Peer review of "The Role of Biomarkers in the Management of Colorectal Liver Metastases"

_cancers, 2022, doi:10.3390/cancers14194602_

Round 1

Reviewer 1 Report (Previous Reviewer 2)

The authors answered all the questions and concerns raised in the first round. I have no further questions.

Reviewer 2 Report (New Reviewer)

I have read the revised manuscript of Hewitt et al. and assessed their responses to the reviewers.  Overall, I consider that all comments and suggestions raised by the reviewers have been fully addressed.

This manuscript is a resubmission of an earlier submission. The following is a list of the peer review reports and author responses from that submission.

Round 1

Reviewer 1 Report

Colorectal cancer is one of the most prominent causes of cancer-related morbidity and death globally. The 5-year overall survival rate for individuals with colorectal liver metastases has risen beyond 50% because to surgical care and better systemic medications (CRLM). Patients with unresectable CRLM have also benefited from an abundance of liver-directed treatments that have enhanced local disease management. 

Whenever metastatic spread occurs, the liver is usually the first place it happens. Patients with advanced colorectal cancer have access to a wide variety of treatment options. However, trustworthy biomarkers to aid with clinical decision-making are lacking. Recent developments in genome sequencing technology have drastically lowered the cost and shortened the time required for research into the causes of colorectal cancer. In this article, the authors explore the current relevance of biomarkers in the therapy of colorectal cancer liver metastases. Given the multitude of therapy modalities and lack of evidence on timing and sequence, reliable biomarkers are required to aid doctors with the formulation of patient-tailored care strategies. Insights from these studies have uncovered biomarkers that may one day help doctors make more educated choices about how to treat patients with CRLM. 

I think it is a very interesting work, but the review of the multi-omics content is still not comprehensive enough and lacks the latest research progress. Therefore, I think the authors should re-review the multi-omics content separately in a more detailed way.

Author Response

  1. I think it is a very interesting work, but the review of the multi-omics content is still not comprehensive enough and lacks the latest research progress. Therefore, I think the authors should re-review the multi-omics content separately in a more detailed way.

We thank you for your constructive comments. A significant amount of multiomic content was modified and added to the molecular biomarkers section (Pages 4-9, lines 150-350) for a more comprehensive discussion.

Thank you for considering our revised manuscript.

Sincerely,

Timothy M. Pawlik, MD, MPH, MBA, MTS, PhD, FACS, FRACS (Hon.)

Professor and Chair, Department of Surgery

The Urban Meyer III and Shelley Meyer Chair for Cancer Research

Professor of Surgery, Oncology, and Health Services Management and Policy

Surgeon in Chief, The Ohio State University Wexner Medical Center

The Ohio State University, Wexner Medical Center

Reviewer 2 Report

The authors reviewed the current role of molecular biomarkers in the management of colorectal liver metastases (CRLM). Several key clinical trials and publications have been included in the current manuscript, RAS, BRAF, HER2 and other well-known markers are listed, but the novel parts such as immune-related genes or other gene signatures created to help CRLM diagnosis are not mentioned. It would be interesting to include those parts to reinforce the role of biomarkers in CRLM. Here are some comments and questions:

Majors:

·       Could AI contribute to the tumor morphology diagnosis and management or radiomics?

·       Adequate contents, maybe an extra section is needed to support Figure 1, otherwise it could only bring confusion to readers.

·       For “molecular biomarkers”, why only include gene mutations? As mentioned in line145-148, “protein and post-translational protein modifications present in metastatic CRC tumors especially in proteins associated with the extracellular matrix, energy metabolism, and immune-cell-related migration”. What is the role of protein modifications, metabolism and immune cell-related genes in CRLM?

Minors:

All the citations in the figure or table legend should go to reference section.

Some sentences are hard to follow like (line 325) “…. are also often based on ….”  

Author Response

The authors reviewed the current role of molecular biomarkers in the management of colorectal liver metastases (CRLM). Several key clinical trials and publications have been included in the current manuscript, RAS, BRAF, HER2 and other well-known markers are listed, but the novel parts such as immune-related genes or other gene signatures created to help CRLM diagnosis are not mentioned. It would be interesting to include those parts to reinforce the role of biomarkers in CRLM. Here are some comments and questions:

  1. Could AI contribute to the tumor morphology diagnosis and management or radiomics?

Thank you for the constructive comments. Given the escalating relevance of AI in medical care, we added additional content on AI and tumor morphology (Page 3, lines 115-123) and radiomics (Page 13, lines 454-465). We also added a recently published AI-based study that explored appropriate surgical margins in KRAS-variant CRLM (Page 7, lines 228-233)

  1. Adequate contents, maybe an extra section is needed to support Figure 1, otherwise it could only bring confusion to readers.

Additional content was added to support Figure 1 (Page 4, lines 158-161)

  1. For “molecular biomarkers”, why only include gene mutations? As mentioned in line145-148, “protein and post-translational protein modifications present in metastatic CRC tumors especially in proteins associated with the extracellular matrix, energy metabolism, and immune-cell-related migration”. What is the role of protein modifications, metabolism and immune cell-related genes in CRLM?

We agree that the discussion surrounding multiomics was incomplete. We have added a significant amount of content to the molecular biomarker section (Pages 4-9, lines 150-350)

  1. All the citations in the figure or table legend should go to reference section.

The citations in the tables/figures have been added to the reference section. Open access citations were left in the caption.

  1. Some sentences are hard to follow like (line 325) “…. are also often based on ….”  

Multiple sentences throughout the text were revised to improve clarity (Page 2, lines 87-93; Page 7, lines 237-247; Page 10, lines 401-408; Pages 12-14, lines 418-451, 466-480, and 494-505).

Thank you for considering our revised manuscript.

Sincerely,

Timothy M. Pawlik, MD, MPH, MBA, MTS, PhD, FACS, FRACS (Hon.)

Professor and Chair, Department of Surgery

The Urban Meyer III and Shelley Meyer Chair for Cancer Research

Professor of Surgery, Oncology, and Health Services Management and Policy

Surgeon in Chief, The Ohio State University Wexner Medical Center

The Ohio State University, Wexner Medical Center